# Efficient Loss-Based Decoding on Graphs for Extreme Classification

**Itay Evron**
Computer Science Dept.
The Technion, Israel
evron.itay@gmail.com

**Edward Moroshko**
Electrical Engineering Dept.
The Technion, Israel
edward.moroshko@gmail.com

**Koby Crammer**
Electrical Engineering Dept.
The Technion, Israel
koby@ee.technion.ac.il

## Abstract

In *extreme classification* problems, learning algorithms are required to map instances to labels from an extremely large label set. We build on a recent extreme classification framework with logarithmic time and space [19], and on a general approach for error correcting output coding (ECOC) with loss-based decoding [1], and introduce a flexible and efficient approach accompanied by theoretical bounds. Our framework employs output codes induced by graphs, for which we show how to perform efficient loss-based decoding to potentially improve accuracy. In addition, our framework offers a tradeoff between accuracy, model size and prediction time. We show how to find the sweet spot of this tradeoff using only the training data. Our experimental study demonstrates the validity of our assumptions and claims, and shows that our method is competitive with state-of-the-art algorithms.

## 1 Introduction

Multiclass classification is the task of assigning instances with a category or class from a finite set. Its numerous applications range from finding a topic of a news item, via classifying objects in images, via spoken words detection, to predicting the next word in a sentence. Our ability to solve multiclass problems with larger and larger sets improves with computation power. Recent research focuses on *extreme classification* where the number of possible classes $K$ is extremely large.

In such cases, previously developed methods, such as *One-vs-One (OVO)* [17], *One-vs-Rest (OVR)* [9] and multiclass SVMs [34, 6, 11, 25], that scale linearly in the number of classes $K$, are not feasible. These methods maintain too large models, that cannot be stored easily. Moreover, their training and inference times are at least linear in $K$, and thus do not scale for extreme classification problems.

Recently, Jasinska and Karampatziakis [19] proposed a Log-Time Log-Space (LTLS) approach, representing classes as paths on graphs. LTLS is very efficient, but has a limited representation, resulting in an inferior accuracy compared to other methods. More than a decade earlier, Allwein et al. [1] presented a unified view of error correcting output coding (ECOC) for classification, as well as the loss-based decoding framework. They showed its superiority over Hamming decoding, both theoretically and empirically.

In this work we build on these two works and introduce an *efficient* (i.e. $O(\log K)$ time and space) loss-based learning and decoding algorithm for *any* loss function of the binary learners' margin. We show that LTLS can be seen as a special case of ECOC. We also make a more general connection between loss-based decoding and graph-based representations and inference. Based on the theoretical framework and analysis derived by [1] for loss-based decoding, we gain insights on how to improve on the specific graphs proposed in LTLS by using more general trellis graphs – which we name *Wide-LTLS (W-LTLS)*. Our method profits from the best of both worlds: better accuracy as in loss-based decoding, and the logarithmic time and space of LTLS. Our empirical study suggests that by

employing coding matrices induced by different trellis graphs, our method allows tradeoffs between accuracy, model size, and inference time, especially appealing for extreme classification.

## 2 Problem setting

We consider *multiclass classification* with $K$ classes, where $K$ is very large. Given a training set of $m$ examples $(x_i, y_i)$ for $x_i \in \mathcal{X} \subseteq \mathbb{R}^d$ and $y_i \in \mathcal{Y} = \{1, ..., K\}$ our goal is to learn a mapping from $\mathcal{X}$ to $\mathcal{Y}$. We focus on the 0/1 loss and evaluate the performance of the learned mapping by measuring its accuracy on a test set – i.e. the fraction of instances with a correct prediction. Formally, the accuracy of a mapping $h : \mathcal{X} \rightarrow \mathcal{Y}$ on a set of $n$ pairs, $\{(x_i, y_i)\}_{i=1}^n$, is defined as $\frac{1}{n} \sum_{i=1}^n \mathbf{1}_{h(x_i)=y_i}$, where $\mathbf{1}_z$ equals 1 if the predicate $z$ is true, and 0 otherwise.

## 3 Error Correcting Output Coding (ECOC)

Dieterich and Bakiri [14] employed ideas from coding theory [23] to create *Error Correcting Output Coding (ECOC)* – a reduction from a multiclass classification problem to multiple binary classification subproblems. In this scheme, each class is assigned with a (distinct) binary codeword of $\ell$ bits (with values in $\{-1, +1\}$). The $K$ codewords create a matrix $M \in \{-1, +1\}^{K \times \ell}$ whose rows are the codewords and whose columns induce $\ell$ partitions of the classes into two subsets. Each of these partitions induces a binary classification subproblem. We denote by $M_k$ the $k$th row of the matrix, and by $M_{k,j}$ its $(k, j)$ entry. In the j$th$ partition, class $k$ is assigned with the binary label $M_{k,j}$.

ECOC introduces redundancy in order to acquire error-correcting capabilities such as a minimum Hamming distance between codewords. The Hamming distance between two codewords $M_a, M_b$ is defined as $\rho(a, b) \triangleq \sum_{j=1}^\ell \frac{1 - M_{a,j} M_{b,j}}{2}$, and the minimum Hamming distance of $M$ is $\rho = \min_{a \neq b} \rho(a, b)$. A high minimum distance of the coding matrix potentially allows overcoming binary classification errors during inference time.

At training time, this scheme generates $\ell$ binary classification training sets of the form $\{x_i, M_{y_i, j}\}_{i=1}^m$ for $j = 1, \ldots, \ell$, and executes some binary classification learning algorithm that returns $\ell$ classifiers, each trained on one of these sets. We assume these classifiers are margin-based, that is, each classifier is a real-valued function, $f_j : \mathcal{X} \rightarrow \mathbb{R}$, whose binary prediction for an input $x$ is $\text{sign}(f_j(x))$. The binary classification learning algorithm defines a margin-based loss $L : \mathbb{R} \rightarrow \mathbb{R}_+$, and minimizes the average loss over the induced set. Formally, $f_j = \arg\min_{f \in \mathcal{F}} \frac{1}{m} \sum_{i=1}^m L(M_{y_i, j} f(x_i))$, where $\mathcal{F}$ is a class of functions, such as the class of bounded linear functions. Few well known loss functions are the hinge loss $L(z) \triangleq \max(0, 1 - z)$, used by SVM, its square, the log loss $L(z) \triangleq \log(1 + e^{-z})$ used in logistic regression, and the exponential loss $L(z) \triangleq e^{-z}$ used in AdaBoost [30].

Once these classifiers are trained, a straightforward inference is performed. Given an input $x$, the algorithm first applies the $\ell$ functions on $x$ and computes a $\{\pm 1\}$-vector of size $\ell$, that is $(\text{sign}(f_1(x)) \ldots \text{sign}(f_\ell(x)))$. Then, the class $k$ which is assigned to the codeword closest in Hamming distance to this vector is returned. This inference scheme is often called *Hamming decoding*.

The Hamming decoding uses only the binary prediction of the binary learners, ignoring the confidence each learner has in its prediction per input. Allwein et al. [1] showed that this margin or confidence holds valuable information for predicting a class $y \in \mathcal{Y}$, and proposed the *loss-based decoding* framework for ECOC[1]. In loss-based decoding, the margin is incorporated via the loss function $L(z)$. Specifically, the class predicted is the one minimizing the total loss

$$k^* = \arg\min_k \sum_{j=1}^\ell L(M_{k,j} f_j(x)) . \tag{1}$$

They [1] also developed error bounds and showed theoretically and empirically that loss-based decoding outperforms Hamming decoding.

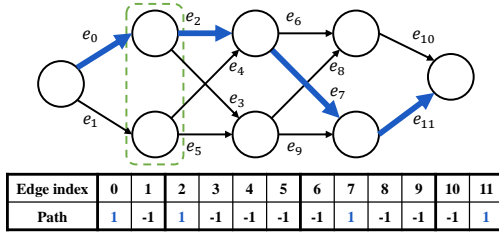

| Edge index | 0 | 1 | 2 | 3 | 4 | 5 | 6 | 7 | 8 | 9 | 10 | 11 |
|---|---|---|---|---|---|---|---|---|---|---|---|---|
| Path | 1 | -1 | 1 | -1 | -1 | -1 | -1 | 1 | -1 | -1 | -1 | 1 |

Figure 1: Path codeword representation. An entry containing 1 means that the corresponding edge is a part of the illustrated bold blue path. The green dashed rectangle shows a vertical slice.

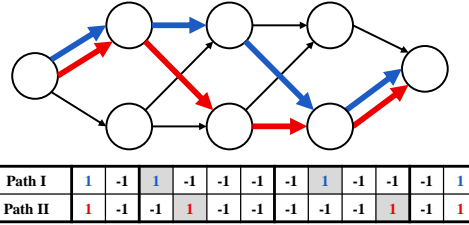

| Path I | 1 | -1 | 1 | -1 | -1 | -1 | -1 | 1 | -1 | -1 | -1 | 1 |
| Path II | 1 | -1 | -1 | 1 | -1 | -1 | -1 | -1 | -1 | 1 | -1 | 1 |

Figure 2: Two closest paths. Predicting Path II (red) instead of I (blue), will result in a prediction error. The Hamming distance between the corresponding codewords is 4. The highlighted entries correspond to the 4 disagreement edges.

One drawback of their method is that given a loss function $L$, loss-based decoding requires an exhaustive evaluation of the total loss for each codeword $M_k$ (each row of the coding matrix). This implies a decoding time at least linear in $K$, making it intractable for extreme classification. We address this problem below.

## 4   LTLS

A recent extreme classification approach, proposed by Jasinska and Karampatziakis [19], performs training and inference in time and space logarithmic in $K$, by embedding the $K$ classes into $K$ paths of a directed-acyclic trellis graph $T$, built compactly with $\ell = \mathcal{O}(\log K)$ edges. We denote the set of vertices $V$ and set of edges $E$. A multiclass model is defined using $\ell$ functions from the feature space to the reals, $w_j(x)$, one function per edge in $E = \{e_j\}_{j=1}^{\ell}$. Given an input, the algorithm assigns weights to the edges, and computes the *heaviest path* using the Viterbi [32] algorithm in $\mathcal{O}(|E|) = \mathcal{O}(\log K)$ time. It then outputs the class (from $\mathcal{Y}$) assigned to the heaviest path.

Jasinska and Karampatziakis [19] proposed to train the model in an online manner. The algorithm maintains $\ell$ functions $f_j(x)$ and works in rounds. In each training round a specific input-output pair $(x_i, y_i)$ is considered, the algorithm performs inference using the $\ell$ functions to predict a class $\hat{y}_i$, and the functions $f_j(x)$ are modified to improve the overall prediction for $x_i$ according to $y_i, \hat{y}_i$. The inference performed during train and test times, includes using the obtained functions $f_j(x)$ to compute the weights $w_j(x)$ of each input, by simply setting $w_j(x) = f_j(x)$. Specifically, they used margin-based learning algorithms, where $f_j(x)$ is the margin of a binary prediction.

Our first contribution is the observation that the LTLS approach can be thought of as an ECOC scheme, in which the codewords (rows) represent paths in the trellis graph, and the columns correspond to edges on the graph. Figure 1 illustrates how a codeword corresponds to a path on the graph.

It might seem like this approach can represent only numbers of classes $K$ which are powers of 2. However, in Appendix C.1 we show how to create trellis graphs with exactly $K$ paths, for any $K \in \mathbb{N}$.

### 4.1   Path assignment

LTLS requires a bijective mapping between paths to classes and vice versa. It was proposed in [19] to employ a greedy assignment policy suitable for online learning, where during training, a sample whose class is yet unassigned with a path, is assigned with the heaviest unassigned path. One could also consider a naive random assignment between paths and classes.

### 4.2   Limitations

The elegant LTLS construction suffers from two limitations:

1. **Difficult induced binary subproblems**: The induced binary subproblems are hard, especially when learned with *linear* classifiers. Each path uses one of four edges between every two adjacent vertical slices. Therefore, each edge is used by $\frac{1}{4}$ of the classes, inducing a $\frac{1}{4}K$-*vs*-$\frac{3}{4}K$ subproblem.

Similarly, the edges connected to the source or sink induce $\frac{1}{2}K$-*vs*-$\frac{1}{2}K$ subproblems. In both cases classes are split into two groups, almost arbitrarily, with no clear semantic interpretation for that partition. For comparison, in 1-vs-Rest (OVR) the induced subproblems are considered much simpler as they require classifying only one class vs the rest[2] (meaning they are much less *balanced*).

2. **Low minimum distance**: In the LTLS trellis architecture, every path has another (closest) path within 2 edge deletions and 2 edge insertions (see Figure 2). Thus, the *minimum Hamming distance* in the underlying coding matrix is restrictively small: $\rho = 4$, which might imply a poor error correcting capability. The OVR coding matrix also suffers from a small minimum distance ($\rho = 2$), but as we explained, the induced subproblems are very simple, allowing a higher classification accuracy in many cases.

We focus on improving the multiclass accuracy by tackling the first limitation, namely making the underlying binary subproblems easier. Addressing the second limitation is deferred to future work.

## 5    Efficient loss-based decoding

We now introduce another contribution – a new algorithm performing efficient loss-based decoding (inference) for *any* loss function by exploiting the structure of trellis graphs. Similarly to [19], our decoding algorithm performs inference in two steps. First, it assigns (per input $x$ to be classified) weights $\{w_j(x)\}_{j=1}^{\ell}$ to the edges $\{e_j\}_{j=1}^{\ell}$ of the trellis graph. Second, it finds the *shortest* path (instead of the heaviest) $P_{k^*}$ by an efficient dynamic programming (Viterbi) algorithm and predicts the class $k^*$. Unlike [19], our strategy for assigning edge weights ensures that for any class $k$, the weight of the path assigned to this class, $w(P_k) \triangleq \sum_{j:e_j \in P_k} w_j(x)$, equals the total loss $\sum_{j=1}^{\ell} L(M_{k,j} f_j(x))$ for the classified input $x$. Therefore, finding the shortest path on the graph is equivalent to minimizing the total loss, which is the aim in loss-based decoding. In other words, we design a new weighting scheme that links loss-based decoding to the shortest path in a graph.

We now describe our algorithm in more detail for the case when the number of classes $K$ is a power of 2 (see Appendix C.2 for extension to arbitrary $K$). Consider a directed edge $e_j \in E$ and denote by $(u_j, v_j)$ the two vertices it connects. Denote by $S(e_j)$ the set of edges outgoing from the same vertical slice as $e_j$. Formally, $S(e_j) = \{(u, u') : \delta(u) = \delta(u_j)\}$, where $\delta(v)$ is the shortest distance from the source vertex to $v$ (in terms of number of edges). For example, in Figure 1, $S(e_0) = S(e_1) = \{e_0, e_1\}$, $S(e_2) = S(e_3) = S(e_4) = S(e_5) = \{e_2, e_3, e_4, e_5\}$. Given a loss function $L(z)$ and an input instance $x$, we set the weight $w_j$ for edge $e_j$ as following,

$$w_j(x) = L(1 \times f_j(x)) + \sum_{j':e_{j'} \in S(e_j) \setminus \{e_j\}} L((-1) \times f_{j'}(x)) . \tag{2}$$

For example, in Figure 1 we have,

$$w_0(x) = L(1 \times f_0(x)) + L((-1) \times f_1(x))$$
$$w_2(x) = L(1 \times f_2(x)) + L((-1) \times f_3(x)) + L((-1) \times f_4(x)) + L((-1) \times f_5(x)) .$$

The next theorem states that for our choice of weights, finding the shortest path in the weighted graph is equivalent to loss-based decoding. Thus, algorithmically we can enjoy fast decoding (i.e. inference), and statistically we can enjoy better performance by using loss-based decoding.

**Theorem 1** *Let $L(z)$ be any loss function of the margin. Let $T$ be a trellis graph with an underlying coding matrix $M$. Assume that for any $x \in \mathcal{X}$ the edge weights are calculated as in Eq. (2). Then, the weight of any path $P_k$ equals to the loss suffered by predicting its corresponding class $k$, i.e. $w(P_k) = \sum_{j=1}^{\ell} L(M_{k,j} f_j(x))$.*

The proof appears in Appendix A. In the next lemma we claim that LTLS decoding is a special case of loss-based decoding with the squared loss function. See Appendix B for proof.

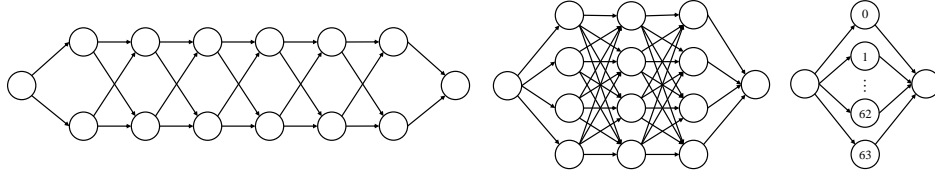

Figure 3: Different graphs for $K = 64$ classes. From left to right: the LTLS graph with a slice width of $b = 2$, W-LTLS with $b = 4$, and the widest W-LTLS graph with $b = 64$, corresponding to OVR.

**Lemma 2** *Denote the squared loss function by $L_{sq}(z) \triangleq (1 - z)^2$. Given a trellis graph represented using a coding matrix $M \in \{-1, +1\}^{K \times \ell}$, and $\ell$ functions $f_j(x)$, for $j = 1 \ldots \ell$, the decoding method of LTLS (mentioned in Section 4) is a special case of loss-based decoding with the squared loss, that is $\arg \max_k w(P_k) = \arg \min_k \left\{ \sum_j L_{sq}(M_{k,j} f_j(x)) \right\}$.*

We next build on the framework of [1] to design graphs with a better multiclass accuracy.

## 6   Wide-LTLS (W-LTLS)

Allwein et al. [1] derived error bounds for loss-based decoding with any convex loss function $L$. They showed that the training multiclass error with loss-based decoding is upper bounded by:

$$\frac{\ell \times \varepsilon}{\rho \times L(0)} \qquad (3)$$

where $\rho$ is the minimum Hamming distance of the code and

$$\varepsilon = \frac{1}{m\ell} \sum_{i=1}^{m} \sum_{j=1}^{\ell} L\left(M_{y_i,j} f_j(x_i)\right) \qquad (4)$$

is the average binary loss on the training set of the learned functions $\{f_j\}_{j=1}^{\ell}$ with respect to a coding matrix $M$ and a loss $L$. One approach to reduce the bound, and thus hopefully also the multiclass training error (and under some conditions also the test error) is to reduce the *total error* of the binary problems $\ell \times \varepsilon$. We now show how to achieve this by generalizing the LTLS framework to a more flexible architecture which we call W-LTLS [3].

Motivated by the error bound of [1], we propose a generalization of the LTLS model. By increasing the *slice width* of the trellis graph, and consequently increasing the number of edges between adjacent vertical slices, the induced subproblems become less balanced and potentially easier to learn (see Remark 2). For simplicity we choose a fixed slice width $b \in \{2, \ldots, K\}$ for the entire graph (e.g. see Figure 3). In such a graph, most of the induced subproblems are $\frac{1}{b^2} K$-*vs-rest* (corresponding to edges between adjacent slices) and some are $\frac{1}{b} K$-*vs-rest* (the ones connected to the source or to the sink). As $b$ increases, the graph representation becomes less compact and requires more edges, i.e. $\ell$ increases. However, the induced subproblems potentially become easier, improving the multiclass accuracy. This suggests that our model allows an *accuracy vs model size tradeoff*.

In the special case where $b = K$ we get the widest graph containing $2K$ edges (see Figure 3). All the subproblems are now 1-*vs-rest*: the $k$th path from the source to the sink contains two edges (one from the source and one to the sink) which are not a part of any other path. Thus, the corresponding two columns in the underlying coding matrix are identical – having 1 at their $k$th entry and $(-1)$ at the rest. This implies that the distinct columns of the matrix could be rearranged as the diagonal coding matrix corresponding to OVR, making our model when $b = K$ an implementation of OVR.

In Section 7 we show empirically that W-LTLS improves the multiclass accuracy of LTLS. In Appendix E.2 we show that the binary subproblems indeed become easier, i.e. we observe a decrease in the average binary loss $\varepsilon$, lowering the bound in (3). Note that the denominator $\rho \times L(0)$ is left untouched – the minimum distance of the coding matrices corresponding to different architectures of W-LTLS is still 4, like in the original LTLS model (see Section 4.2).

## 6.1 Time and space complexity analysis

W-LTLS requires training and storing a binary learner for every edge. For most linear classifiers (with $d$ parameters each) we get[4] a total *model size complexity* and an *inference time complexity* of $\mathcal{O}\left(d\left|E\right|\right) = \mathcal{O}\left(d\frac{b^2}{\log b}\log K\right)$ (see Appendix D for further details). Moreover, many extreme classification datasets are sparse – the average number of non-zero features in a sample is $d_e \ll d$. The inference time complexity thus decreases to $\mathcal{O}\left(d_e\frac{b^2}{\log b}\log K\right)$.

This is a significant advantage: while inference with loss-based decoding for general matrices requires $\mathcal{O}\left(d_e\ell + K\ell\right)$ time, our model performs it in only $\mathcal{O}\left(d_e\ell + \ell\right) = \mathcal{O}\left(d_e\ell\right)$.

Since training requires learning $\ell$ binary subproblems, the *training time complexity* is also sublinear in $K$. These subproblems can be learned separately on $\ell$ cores, leading to major speedups.

## 6.2 Wider graphs induce sparse models

The high sparsity typical to extreme classification datasets (e.g. the `Dmoz` dataset has $d = 833,484$ features, but on average only $d_e = 174$ of them are non-zero), is heavily exploited by previous works such as PD-Sparse [15], PPDSparse [35], and DiSMEC [2], which all learn sparse models.

Indeed, we find that for sparse datasets, our algorithm typically learns a model with a low percentage of non-zero weights. Moreover, the percentage of non-zero decreases significantly as the slice width $b$ is increased (see Appendix E.6). This allows us to employ a simple post-pruning of the learned weights. For some threshold value $\lambda$, we set to zero all learned weights in $[-\lambda, \lambda]$, yielding a sparse model. Similar approaches were taken by [2, 19, 15] either explicitly or implicitly.

In Section 7.3 we show that the above scheme successfully yields highly sparse models.

# 7 Experiments

We test our algorithms on 5 extreme multiclass datasets previously used in [15], having approximately $10^2$, $10^3$, and $10^4$ classes (see Table 1 in Appendix E.1). We use AROW [10] to train the binary functions $\{f_j\}_{j=1}^{\ell}$ of W-LTLS. Its online updates are based on the squared hinge loss $L_{SH}(z) \triangleq \left(\max\left(0, 1-z\right)\right)^2$. For each dataset, we build wide graphs with multiple slice widths. For each configuration (dataset and graph) we perform five runs using random sample shuffling on every epoch, and a random path assignment (as explained in Section 4.1, unlike the greedy policy used in [19]), and report averages over these five runs. Unlike [19], we train the $\ell$ binary learners *independently* rather than in a joint (structured) manner. This allows parallel independent training, as common for training binary learners for ECOC, with no need to perform full multiclass inference during training.

## 7.1 Loss-based decoding

We run W-LTLS with different loss functions for loss-based decoding: the exponential loss, the squared loss (used by LTLS, see Lemma 2), the log loss, the hinge loss, and the squared hinge loss.

The results appear in Figure 4. We observe that decoding with the exponential loss works the best on all five datasets. For the two largest datasets (`Dmoz` and `LSHTC1`) we report significant accuracy improvement when using the exponential loss for decoding in graphs with large slice widths ($b$), over the squared loss used implicitly by LTLS. Indeed, for these larger values of $b$, the subproblems are easier (see Appendix E.2 for detailed analysis). This should result in larger prediction margins $|f_j(x)|$, as we indeed observe empirically (shown in Appendix E.4). The various loss functions $L(z)$ differ significantly for $z \ll 0$, potentially explaining why we find larger accuracy differences as $b$ increases when decoding with different loss functions.

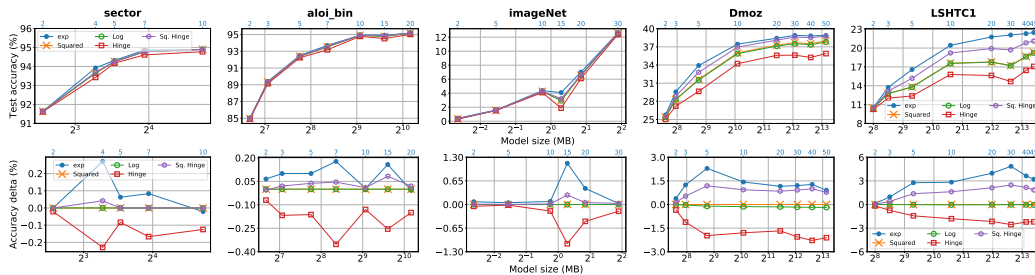

Figure 4: First row: Multiclass test accuracy as a function of the model size (MBytes) for loss-based decoding with different loss functions. Second row: Relative increase in multiclass test accuracy compared to decoding with the squared loss used implicitly in LTLS. The secondary x-axes (top axes, blue) indicate the slice widths (*b*) used for the W-LTLS trellis graphs.

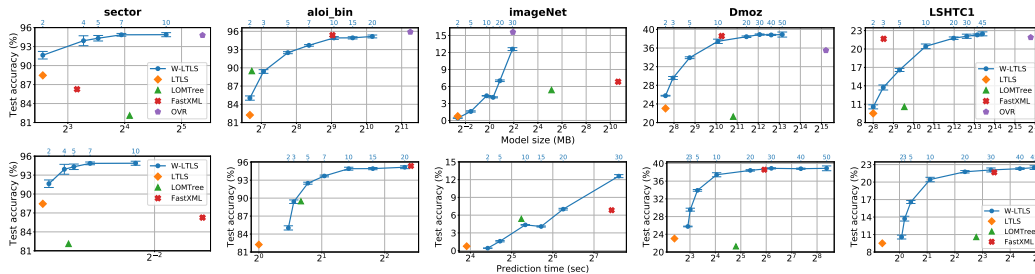

Figure 5: First row: Multiclass test accuracy vs model size. Second row: Multiclass test accuracy vs prediction time. A 95% confidence interval is shown for the results of W-LTLS.

## 7.2 Multiclass test accuracy

We compare the multiclass test accuracy of W-LTLS (using the exponential loss for decoding) to the same baselines presented in [19]. Namely we compare to LTLS [19], LOMTree [7] (results quoted from [19]), FastXML [29] (run with the default parameters on the same computer as our model), and OVR (binary learners trained using AROW). For convenience, the results are also presented in a tabular form in Appendix E.5.

### 7.2.1 Accuracy vs Model size

The first row of Figure 5 (best seen in color) summarizes the multiclass accuracies vs model size.

Among the four competitors, LTLS enjoys the smallest model size, LOMTree and FastXML have larger model sizes, and OVR is the largest. LTLS achieves lower accuracies than LOMTree on two datasets, and higher ones on the other two. OVR enjoys the best accuracy, yet with a price of model size. For example, in Dmoz, LTLS achieves $23\%$ accuracy vs $35.5\%$ of OVR, though the model size of the latter is $\times 200$ larger than of the former.

In all five datasets, an increase in the slice width of W-LTLS (and consequently in the model size) translates almost always to an increase in accuracy. Our model is often better or competitive with the other algorithms that have logarithmic inference time complexity (LTLS, LOMTree, FastXML), and also competitive with OVR in terms of accuracy, while we still enjoy much smaller model sizes.

For the smallest model sizes of W-LTLS (corresponding to $b = 2$), our trellis graph falls back to the one of LTLS. The accuracies gaps between these two models may be explained by the different binary learners the experiments were run with – LTLS used averaged Perceptron as the binary learner whilst we used AROW. Also, LTLS was trained in a structured manner with a greedy path assignment policy while we trained every binary function independently with a random path assignment policy (see Section 6.1). In our runs we observed that independent training achieves accuracy competitive with to structured online training, while usually converging much faster. It is interesting to note that for

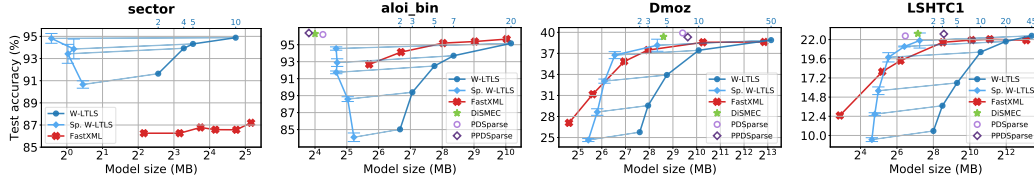

Figure 6: Multiclass test accuracy vs model size for sparse models. Lines between two W-LTLS plots connect the same models before and after the pruning. The secondary x-axes (top axes, blue) indicate the slice widths (*b*) used for the (unpruned) W-LTLS trellis graphs.

the `imageNet` dataset the LTLS model cannot fit the data, i.e the training error is close to 1 and the test accuracy is close to 0. The reason is that the binary subproblems are very hard, as was also noted by [19]. By increasing the slice width (*b*), the W-LTLS model mitigates this underfitting problem, still with logarithmic time and space complexity.

We also observe in the first row of Figure 5 that there is a point where the multiclass test accuracy of W-LTLS starts to saturate (except for `imageNet`). Our experiments show that this point can be found by looking at the training error and its bound only. We thus have an effective way to choose the optimal model size for the dataset and space/time budget at hand by performing model selection (width of the graph in our case) using the training error bound only (see detailed analysis Appendix E.2 and Appendix E.3).

### 7.2.2 Accuracy vs Prediction time

In the second row of Figure 5 we compare prediction (inference) time of W-LTLS to other methods. LTLS enjoys the fastest prediction time, but suffers from low accuracy. LOMTree runs slower than LTLS, but sometimes achieves better accuracy. Despite being implemented in Python, W-LTLS is competitive with FastXML, which is implemented in C++.

### 7.3 Exploiting the sparsity of the datasets

We now demonstrate that the post-pruning proposed in Section 6.2, which zeroes the weights in $[-\lambda, \lambda]$, is highly beneficial. Since `imageNet` is not sparse at all, we do not consider it in this section.

We tune the threshold $\lambda$ so that the degradation in the multiclass validation accuracy is at most $1\%$ (tuning the threshold is done after the cumbersome learning of the weights, and does not require much time).

In Figure 6 we plot the multiclass test accuracy versus model size for the non-sparse W-LTLS, as well as the sparse W-LTLS after pruning the weights as explained above. We compare ourselves to the aforementioned sparse competitors: DiSMEC, PD-Sparse, and PPDSparse (all results quoted from [35]). Since the aforementioned FastXML [29] also exploits sparsity to reduce the size of the learned trees, we consider it here as well (we run the code supplied by the authors for various numbers of trees). For convenience, all the results are also presented in a tabular form in Appendix E.6.

We observe that our method can induce very sparse binary learners with a small degradation in accuracy. In addition, as expected, the wider the graphs (larger *b*), the more beneficial is the pruning. Interestingly, while the number of parameters increases as the graphs become wider, the actual storage space for the pruned sparse models may even decrease. This phenomenon is observed for the `sector` and `aloi.bin` datasets.

Finally, we note that although PD-Sparse [15] and DiSMEC [2] perform better on some model size regions of the datasets, their worse case space requirement during training is linear in the number of classes $K$, whereas our approach guarantees (adjustable) logarithmic space for training.

## 8 Related work

Extreme classification was studied extensively in the past decade. It faces unique challenges, amongst which is the model size of its designated learning algorithms. An extremely large model size often

implies long training and test times, as well as excessive space requirements. Also, when the number of classes $K$ is extremely large, the inference time complexity should be sublinear in $K$ for the classifier to be useful.

The *Error Correcting Output Coding (ECOC)* (see Section 3) approach seems promising for extreme classification, as it potentially allows a very compact representation of the label space with $K$ codewords of length $\ell = \mathcal{O}(\log K)$. Indeed, many works concentrated on utilizing ECOC for extreme classification. Some formulate dedicated optimization problems to find ECOC matrices suitable for extreme classification [8] and others focus on learning better binary learners [24].

However, very little attention has been given to the decoding time complexity. In the multiclass regime where only one class is the correct class, many of these works are forced to use exact (i.e. not approximated) decoding algorithms which often require $\mathcal{O}(K\ell)$ time [21] in the worst-case. Norouzi et al. [27] proposed a fast exact search nearest neighbor algorithm in the Hamming space, which for coding matrices suitable for extreme classification can achieve $o(K)$ time complexity, but not $\mathcal{O}(\log K)$. These algorithms are often limited to binary (dense) matrices and hard decoding. Some approaches [22] utilize graphical processing units in order to find the nearest neighbor in Euclidean space, which can be useful for soft decoding, but might be too demanding for weaker devices. In our work we keep the time complexity of any loss-based decoding logarithmic in $K$.

Moreover, most existing ECOC methods employ coding matrices with higher minimum distance $\rho$, but with balanced binary subproblems. In Section 6 we explain how our ability of inducing less balanced subproblems is beneficial both for the learnability of these subproblems, and for the post-pruning of learned weights to create sparse models.

It is also worth mentioning that many of the ECOC-based works (like randomized or learned codes [8, 37]) require storing the entire coding matrix even during inference time. Hence, the additional space complexity needed only for decoding during inference is $\mathcal{O}(K \log K)$, rather than $\mathcal{O}(K)$ as in LTLS and W-LTLS which do not directly use the coding matrix for decoding the binary predictions and only require a mapping from code to label (e.g. a binary tree).

Naturally, hierarchical classification approaches are very popular for extreme classification tasks. Many of these approaches employ tree based models [3, 29, 28, 18, 20, 7, 12, 4, 26, 13]. Such models can be seen as decision trees allowing inference time complexity linear in the tree height, that is $\mathcal{O}(\log K)$ if the tree is (approximately) balanced. A few models even achieve logarithmic training time, e.g. [20]. Despite having a sublinear time complexity, these models require storing $\mathcal{O}(K)$ classifiers.

Another line of research focused on label-embedding methods [5, 31, 33, 36]. These methods try to exploit label correlations and project the labels onto a low-dimensional space, reducing training and prediction time. However, the low-rank assumption usually leads to an accuracy degradation.

Linear methods were also the focus of some recent works [2, 15, 35]. They learn a linear classifier per label and incorporate sparsity assumptions or perform distributed computations. However, the training and prediction complexities of these methods do not scale gracefully to datasets with a very large number of labels. Using a similar post-pruning approach and independent (i.e. not joint) learning of the subproblems, W-LTLS is also capable of exploiting sparsity and learn in parallel.

## 9 Conclusions and Future work

We propose a new efficient loss-based decoding algorithm that works for any loss function. Motivated by a general error bound for loss-based decoding [1], we show how to build on the log-time log-space (LTLS) framework [19] and employ a more general type of trellis graph architectures. Our method offers a tradeoff between multiclass accuracy, model size and prediction time, and achieves better multiclass accuracies under logarithmic time and space guarantees.

Many intriguing directions remain uncovered, suggesting a variety of possible future work. One could try to improve the restrictively low minimum code distance of W-LTLS discussed in Section 4.2 Regularization terms could also be introduced, to try and further improve the learned sparse models. Moreover, it may be interesting to consider weighing every entry of the coding matrix (in the spirit of Escalera et al. [16]) in the context of trellis graphs. Finally, many ideas in this paper can be extended for other types of graphs and graph algorithms.

**Acknowledgements**

We would like to thank Eyal Bairey for the fruitful discussions. This research was supported in part by The Israel Science Foundation, grant No. 2030/16.

## Footnotes

[1] Another contribution of their work, less relevant to our work, is a unifying approach for multiclass classification tasks. They showed that many popular approaches are unified into a framework of sparse (ternary) coding schemes with a coding matrix $M \in \{-1, 0, 1\}^{K \times \ell}$. For example, One-vs-Rest (OVR) could be thought of as $K \times K$ matrix whose diagonal elements are 1, and the rest are -1.

[2] A similar observation is given in Section 6 of Allwein et al. [1] regarding OVR.

[3]Code is available online at https://github.com/ievron/wltls/

[4] Clearly, when $b \approx \sqrt{K}$ our method cannot be regarded as sublinear in $K$ anymore. However, our empirical study shows that high accuracy can be achieved using much smaller values of $b$.

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
