[Supplementary Material]

# SUPPLEMENTARY MATERIAL

## A  Proof of Theorem 1

**Proof:** For any class $k$ we have,

$$w\left(P_k\right) = \sum_{j:e_j \in P_k} w_j\left(x\right) = \sum_{j:e_j \in P_k} \left[L\left(1 \times f_j\left(x\right)\right) + \sum_{j':e_{j'} \in S(e_j)\backslash\{e_j\}} L\left((-1) \times f_{j'}(x)\right)\right]$$

$$= \sum_{j:e_j \in P_k} \left[\sum_{j':e_{j'} \in S(e_j)} L\left(M_{k,j'} \times f_{j'}(x)\right)\right] = \sum_{j=1}^{\ell} L\left(M_{k,j} f_j(x)\right) .$$

The third equality in the proof follows from the path codeword representation, see for example Figure 1. ∎

## B  Proof of Lemma 2

**Proof:** Indeed, for LTLS we have,

$$\arg\max_k w\left(P_k\right) = \arg\max_k \left\{\sum_{j:e_j \in P_k} w_j\left(x\right)\right\} = \arg\max_k \left\{\sum_{j:M_{k,j}=1} f_j\left(x\right)\right\}$$

$$= \arg\max_k \left\{2 \sum_{j:M_{k,j}=1} f_j\left(x\right) - \underbrace{\sum_j f_j\left(x\right)}_{\text{constant in } k}\right\} = \arg\max_k \left\{\sum_{j:M_{k,j}=1} f_j\left(x\right) - \sum_{j:M_{k,j}=-1} f_j\left(x\right)\right\}$$

$$= \arg\max_k \left\{\sum_j M_{k,j} f_j\left(x\right)\right\} = \arg\max_k \left\{\underbrace{\sum_j \cancel{M_{k,j}^2}}_{=\ell} + \underbrace{\sum_j \cancel{f_j^2(x)}}_{\text{constant in } k} - \sum_j \left(M_{k,j} - f_j\left(x\right)\right)^2\right\}$$

$$= \arg\min_k \left\{\sum_j \left(M_{k,j} - f_j\left(x\right)\right)^2 \underbrace{\left(M_{k,j}\right)^2}_{=1}\right\} = \arg\min_k \left\{\sum_j \left(1 - M_{k,j} f_j\left(x\right)\right)^2\right\}$$

$$= \arg\min_k \left\{\sum_j L_{sq}\left(M_{k,j} f_j\left(x\right)\right)\right\} .$$

Note that we used the fact that $M_{k,j} \in \{-1, +1\}$. ∎

## C  Extension to arbitrary K

### C.1  Graph construction

The following algorithm construct graphs with any number of paths, and is not restricted to powers of 2. See Figure 7 for examples.

---

**Algorithm 1:** Graph construction with an arbitrary $K$

---

**Input** :Number of labels $K$ and slice width $b$.

1 Convert $K$ to a base-$b$ representation.
2 Store the *reverse* representation in an array $A$ of size $\lfloor \log_b K \rfloor + 1$ (i.e. the least significant $b$-ary *digit* is in $A\,[0]$).
3 Build a trellis graph with $\lfloor \log_b K \rfloor + 1$ inner slices, each with $b$ vertices.
4 Add a source vertex $s$ and connect it to the vertices of the first inner slice.
5 Add a sink vertex $t$.
6 For every inner slice $i = 0 \ldots \lfloor \log_b K \rfloor$, connect $A\,[i]$ vertices of the slice to the sink.
7 Delete any vertices from which the sink is unreachable.

---

Figure 7: An illustration of the graph construction algorithm for two different values of $K$, using $b = 3$. The darker edges are the ones created on stage 6 of Algorithm 1. Note that in the upper construction, the minimum distance between any two path is actually $\rho = 3$ and not $\rho = 4$ like previously noted. This sometimes holds minor accuracy implications.

Below we show that this construction indeed produces a graph with exactly $K$ paths from source to sink. We start with a technical lemma.

**Lemma 3** *Let $v \in V \setminus \{t\}$ be a vertex in the ith inner slice (where $i \in \{0 \ldots \lfloor \log_b K \rfloor\}$), and let $\mathcal{N}(v)$ be the number of paths from the source to v. Then $\mathcal{N}(v) = b^i$.*

**Proof:**  We show this by induction on the slice index $i = 0 \ldots \lfloor \log_b K \rfloor$. In the base case $i = 0$ and $v$ is in the first inner slice (see Figure 7). It has only one incoming edge from the source $s$, hence the statement holds, $\mathcal{N}(v) = b^0 = 1$.

Next, denote $in(v) \triangleq \{u : \exists\,(u, v) \in E\}$ and suppose the theorem holds for all slices until the $k$th slice. Let $v$ be a vertex in the $(k + 1)$th slice. Following the graph construction, $v$ has $b$ incoming edges from the vertices in $in(v)$, all of which are in the $k$th slice. According to the inductive hypothesis, $\forall u \in in(v) : \mathcal{N}(u) = b^k$. Every path from $s$ to $u \in in(v)$, could be extended to a path from $s$ to $v$ by concatenating the corresponding edge, and so we get that $\mathcal{N}(v) = \sum_{u \in in(v)} \mathcal{N}(u) = \sum_{u \in in(v)} b^k = b \cdot b^k = b^{k+1}$. ∎

**Theorem 4** *The number of paths from the source to the sink is $K$, i.e. $\mathcal{N}(t) = K$.*

**Proof:** Let $A \in \{0 \ldots b-1\}^{\lfloor \log_b K \rfloor + 1}$ be the array of the reverse base-$b$ representation of $K$. Using the decomposition of the $b$-ary representation, i.e. $K = \sum_{i=0}^{\lfloor \log_b K \rfloor} A[i] \cdot b^i$, and Lemma 3, we get:

$$\mathcal{N}(t) = \sum_{v \in in(t)} \mathcal{N}(v) = \sum_{i=0}^{\lfloor \log_b K \rfloor} \left[ \sum_{v \in \text{slice}_i \cap in(t)} \underbrace{\mathcal{N}(v)}_{=b^i} \right] = \sum_{i=0}^{\lfloor \log_b K \rfloor} \underbrace{|\text{slice}_i \cap in(t)|}_{=A[i]} \cdot b^i$$

$$= \sum_{i=0}^{\lfloor \log_b K \rfloor} A[i] \cdot b^i = K \ .$$

∎

### C.2 Loss-based decoding generalization

We now show how to adjust the generalization of loss-based decoding for graphs with an arbitrary number of paths $K$, constructed using Algorithm 1.

The idea of the reduction in (2) is that going through an edge $e_j$ inflicts the loss of turning its corresponding bit on (i.e. $L(1 \times f_j(x))$), but also the loss of turning off the bits corresponding to other edges between its slices (i.e. $\sum_{j':e_{j'} \in S(e_j) \setminus \{e_j\}} L((-1) \times f_{j'}(x))$), which cannot coappear with $e_j$ in the same path (i.e. a codeword).

The only change in the general case is that an edge $e_j = (u_j, t)$ that is connected to the sink $t$ cannot coappear with *any* other edge outgoing from a vertex in the same vertical slice as $u_j$, **or** that is reachable from $u_j$.

Let $\delta(v)$ be the shortest distance from the source to $v$ (in terms of number of edges).

We define an updated $S$ function (which is a generalization of the one defined in Section 5), where for every edge $e_j = (u_j, v_j) \in E$ we set:

$$S(e_j) = \begin{cases} \{(u, u') : \delta(u) = \delta(u_j)\} & v_j \neq t \\ \{(u, u') : \delta(u) \geq \delta(u_j)\} & v_j = t \end{cases}, \tag{5}$$

and the weights are set as in (2) but with the new $S$ function defined in (5),

$$w_j(x) = L(1 \times f_j(x)) + \sum_{j':e_{j'} \in S(e_j) \setminus \{e_j\}} L((-1) \times f_{j'}(x)) \ . \tag{6}$$

For example, in the following figure we have,

$$S(e_6) = \{e_6, e_7, e_8, e_9\} \qquad S(e_{11}) = \{e_{10}, e_{11}\} \qquad S(e_{13}) = \{e_2, \ldots, e_{13}\} \ .$$
$$S(e_2) = \{e_2, e_3, e_4, e_5, e_{13}\} \qquad S(e_{12}) = \{e_{12}\}$$

Figure 8: An illustration of a graph with $K = 9$ and $b = 2$.

Below we show that Theorem 1 is correct for any $K$. Let $P$ be a path on the trellis graph from the source $s$ to the sink $t$. We start will the following lemma.

**Lemma 5** *Let $e_q = (u_q, t)$ be the last edge in $P$. For every edge $e_j = (u_j, v_j) \in P \setminus \{e_q\}$ we have $\delta(v_j) = \delta(u_j) + 1$.*

**Proof:** Following immediately from the graph construction – other than edges to the sink, there are only edges between adjacent slices (without cycles). Therefore, any vertex $v$ in a vertical slice $i$ (where the leftmost slice containing $s$ is the first one, i.e. $i = 0$) holds $\delta(v) = i$, and every edge $e_j = (u_j, v_j) \in P \setminus \{e_q\}$ holds $\delta(v_j) = \delta(u_j) + 1$. ∎

The next corollaries follow immediately.

**Corollary 6** *For every two different vertices $u, v \in V \setminus \{t\}$ along $P$ we have $\delta(u) \neq \delta(v)$.*

**Corollary 7** *Let $e_q = (u_q, t)$ be the last edge in $P$. For every edge $e_j = (u_j, v_j) \in P \setminus \{e_q\}$ we have $\delta(u_j) < \delta(u_q)$.*

Clearly, since the graph is directed and acyclic, we get the next lemma.

**Lemma 8** *Every vertex in $V$ can have at most one incoming edge and one outgoing edge in $P$.*

By using Lemma 8, Corollary 6 and Lemma 5 we have the next corollary.

**Corollary 9** *Set an edge $e_j = (u_j, v_j)$ along the path $P$. Then, $\forall e_{j'} \in S(e_j) \setminus \{e_j\}$ we have $e_{j'} \notin P$.*

**Proof:** Let $e_{j'} = (u_{j'}, v_{j'})$ be any edge in $S(e_j) \setminus \{e_j\}$, and assume $e_{j'} \in P$. We consider two cases:

1. $v_j \neq t$:
   By the definition of $S(e_j)$, we get $\delta(u_{j'}) = \delta(u_j)$. By negating Corollary 6 we get $u_{j'} = u_j$. Therefore $e_j, e_{j'}$ are two edges in $P$ outgoing from the same vertex $u_j$, in contradiction to Lemma 8.

2. $v_j = t$:
   By definition, $e_j \neq e_{j'}$. By Lemma 8 we get that $u_j \neq u_{j'}$. Also, since $t$ is a sink it has no outgoing edges, thus $u_{j'} \neq t$.
   By the definition of $S(e_j)$, we get $\delta(u_{j'}) \geq \delta(u_j)$. Since $u_{j'} \neq u_j$, we can use Corollary 6 to rule out equivalence and get $\delta(u_{j'}) > \delta(u_j)$. Following Lemma 5, $u_{j'}$ must appear later than $u_j$ in $P$. However, $t$ is the only vertex in $P$ to appear after $u_j$ ($t$ is a sink), and since $u_{j'} \neq t$, we get a contradiction.

∎

We conclude with the main result of this section, which states that Theorem 1 is correct for any $K$.

**Theorem 10** *Following the notations of Theorem 1, assume the weights of the edges are calculated as in Eq. (6) with the $S$ function defined in (5). Then, the weight of any path $P_k$ corresponding to class $k$ equals to the loss suffered by predicting class $k$, i.e. $w(P_k) = \sum_{j=1}^{\ell} L\left(M_{k,j} f_j(x)\right)$.*

**Proof:**

For any class $k$, we denote the last edge in $P_k$ by $e_q = (u_q, t)$. We have,

$$w\left(P_k\right) = \sum_{j:e_j \in P_k} w_j\left(x\right)$$

$$= \sum_{j:e_j \in P_k} \left[ L\left( \underbrace{1}_{=M_{k,j}} \times f_j\left(x\right) \right) + \sum_{j':e_{j'} \in S(e_j) \setminus \{e_j\}} L\left( \underbrace{(-1)}_{\substack{=M_{k,j'} \\ \text{(Corollary 9)}}} \times f_{j'}(x) \right) \right]$$

$$= \sum_{j:e_j \in P_k} \left[ \sum_{j':e_{j'} \in S(e_j)} L\left(M_{k,j'} \times f_{j'}(x)\right) \right]$$

$$= \sum_{j:e_j \in S(e_q)} L\left(M_{k,j} \times f_j(x)\right) + \sum_{j:e_j \in P_k \setminus \{e_q\}} \left[ \sum_{j':e_{j'} \in S(e_j)} L\left(M_{k,j'} \times f_{j'}(x)\right) \right]$$

$$= \sum_{\substack{j:e_j=(u_j,v_j), \\ \delta(u_j) \geq \delta(u_q)}} L\left(M_{k,j} \times f_j(x)\right) + \underbrace{\sum_{j:e_j=(u_j,v_j) \in P_k \setminus \{e_q\}} \left[ \sum_{\substack{j':e_{j'}=(u_{j'},v_{j'}), \\ \delta(u_{j'})=\delta(u_j)}} L\left(M_{k,j'} \times f_{j'}(x)\right) \right]}_{\text{(Corollary 7)} \quad = \sum_{\substack{j:e_j=(u_j,v_j), \\ \delta(u_j)<\delta(u_q)}} L(M_{k,j} \times f_j(x))}$$

$$= \sum_{\substack{j:e_j=(u_j,v_j), \\ \delta(u_j) \geq \delta(u_q)}} L\left(M_{k,j} \times f_j(x)\right) + \sum_{\substack{j:e_j=(u_j,v_j), \\ \delta(u_j)<\delta(u_q)}} L\left(M_{k,j} \times f_j(x)\right)$$

$$= \sum_{j:e_j \in E} L\left(M_{k,j} \times f_j(x)\right)$$

$$= \sum_{j=1}^{\ell} L\left(M_{k,j} f_j(x)\right) \ .$$

∎

# D   Details for complexity analysis in Section 6.1

W-LTLS requires training and storing a binary function or model for every edge. Hence, we first turn to analyze the number of edges.

**Lemma 11** *The number of vertices in the inner slices is at most* $(\lfloor \log_b K \rfloor + 1) \, b$.

**Proof:** Follows immediately from the construction in Appendix C.1. ∎

**Corollary 12** *The number of edges is upper bounded:*

$$|E| \le (b+1) \left( \lfloor \log_b K \rfloor + 1 \right) b + b = \mathcal{O} \left( \frac{b^2}{\log b} \log K \right) .$$

**Proof:** Each vertex in the inner slices can have at most $b + 1$ outgoing edges. We use Lemma 11 and count also the $b$ edges outgoing from the source. ∎

For most linear classifiers (with $d$ parameters each) we get a total *model size complexity* of $\mathcal{O} \left( d \frac{b^2}{\log b} \log K \right)$.

Inference consists of four steps:

1. Computing the value (margin) of all binary functions on the input $x$. This requires $\mathcal{O}(d \, |E|) = \mathcal{O} \left( d \frac{b^2}{\log b} \log K \right)$ time.

2. Computing the edge weights $\{w_i(x)\}_{i=1}^{\ell}$ as explained in Section 5. This can be performed in $\mathcal{O}(|V| + |E|) = \mathcal{O} \left( \frac{b^2}{\log b} \log K \right)$ time using a simple dynamic programming algorithm (e.g. implementing back recursion).

3. Finding the shortest path in the trellis graph with respect to $\{w_i(x)\}_{i=1}^{\ell}$ using the Viterbi algorithm in $\mathcal{O}(|V| + |E|) = \mathcal{O} \left( \frac{b^2}{\log b} \log K \right)$ time.

4. Decoding the shortest path to a class. As explained in Section 4.1, the inference requires a mapping function from path to code. Using data structures such as a binary tree, this can be performed in a $\mathcal{O}(|E|) = \mathcal{O} \left( \frac{b^2}{\log b} \log K \right)$ time complexity.

We get that the total *inference time complexity* is $\mathcal{O}(d \, |E|) = \mathcal{O} \left( d \frac{b^2}{\log b} \log K \right)$.

# E Experiments appendix

## E.1 Datasets used in experiments

| DATASET | CLASSES $K$ | FEATURES $d$ | TRAIN SAMPLES $m$ |
|---|---|---|---|
| SECTOR | 105 | 55,197 | 7,793 |
| ALOI_BIN | 1,000 | 636,911 | 90,000 |
| IMAGENET | 1,000 | 1,000 | 1,125,264 |
| DMOZ | 11,947 | 833,484 | 335,068 |
| LSHTC1 | 12,294 | 1,199,856 | 83,805 |

Table 1: Datasets used in the experiments.

## E.2 Average binary training loss

We validate our hypothesis that wider graphs lead to easier binary problems. In the first row of Figure 9 we plot the average binary training loss $\varepsilon$ as a function of model size. The average is both over the induced binary subproblems and over the five runs.

In all datasets we observe a decrease of the average error as the slice width $b$ grows. The decrease is sharp for low values of $b$ and then practically almost converges (to zero). These plots validate our claim – as the subproblems become more unbalanced they also become easier.

Figure 9: First row: Average binary loss ($\varepsilon$) vs Model size. Second row: Multiclass training error and multiclass training error bound (on a logarithmic scale) vs Model size. The secondary x-axes (top axes, blue) indicate the slice widths (b) used for the W-LTLS trellis graphs.

## E.3 Multiclass training error

In the second row of Figure 9 we plot the multiclass training error (when using loss-based decoding defined in (1) with the squared hinge loss) and its bound (3) for different model sizes [5] (MBytes).

For the bound, we set the minimum distance $\rho = 4$ as explained in Section 6, and $L_{SH}(0) = 1$. The average binary loss $\varepsilon$ was computed as in (4).

For all datasets, the multiclass training error follows qualitatively its bound. For the two larger datasets, shown in the two right panels, both the error and its bound decrease to some point, and then start to increase. This can be explained as follows: at some point, the increase in the slice widths (and $\ell$ and the model size), stops to significantly decrease $\epsilon$ (see first row of Figure 9), such that the term $\ell \times \varepsilon$ appears in the training error bound (3) overall starts increasing (recall that the denominator $\rho \times L(0)$ is constant). By comparing these plots to the multiclass *test* accuracy plots in Figure 5, we observe that at the same point where the training error and its bound start to increase, the test accuracy does not increase significantly anymore. For example, for LSHTC1 and Dmoz datasets, the training error bounds start to increase at around model size of $2^{12}$, and at the same time the test accuracy stops increasing significantly. This suggests that model size of $2^{12}$ is a good point in terms of accuracy/model size tradeoff.

### E.4 Average predictions margin

As discussed in Section 7.1, the following Figure 10 shows that for larger values of $b$, the predictions margin increases.

Figure 10: The average absolute margin, i.e. $\frac{1}{m\ell} \sum_{i=1}^{m} \sum_{j=1}^{\ell} |f_j(x_i)|$, vs the slice width $b$.

## E.5 Experimental results of the multiclass test accuracy experiments

Following are the results of Section 7.2 organized in a tabular form – the model sizes and prediction times of the tested algorithms and their test accuracy. The results of W-LTLS are averaged on five runs.

| DATASET | ALGORITHM | | MODEL SIZE (BYTES) | PREDICTION TIME (SEC) | TEST ACCURACY (%) |
|---|---|---|---|---|---|
| SECTOR | LTLS | | 5.9 MB | 0.14 | 88.45 |
| | W-LTLS | $b = 2$ | 5.9 MB | 0.14 | 91.63 |
| | | $b = 4$ | 9.7 MB | 0.16 | 93.92 |
| | | $b = 5$ | 11.6 MB | 0.16 | 94.32 |
| | | $b = 7$ | 15.4 MB | 0.18 | 94.86 |
| | | $b = 10$ | 26.5 MB | 0.23 | 94.88 |
| | LOMTREE | | 17.0 MB | 0.16 | 82.10 |
| | FASTXML | | 8.9 MB | 0.32 | 86.26 |
| | OVR | | 41.3 MB | - | 94.79 |
| ALOI.BIN | LTLS | | 102.1 MB | 1.0 | 82.24 |
| | W-LTLS | $b = 2$ | 102.1 MB | 1.4 | 85.03 |
| | | $b = 3$ | 133.6 MB | 1.5 | 89.39 |
| | | $b = 5$ | 216.2 MB | 1.7 | 92.49 |
| | | $b = 7$ | 328.0 MB | 2.1 | 93.70 |
| | | $b = 10$ | 537.0 MB | 2.7 | 94.92 |
| | | $b = 15$ | 777.5 MB | 3.5 | 94.93 |
| | | $b = 20$ | 1.1 GB | 5.0 | 95.16 |
| | LOMTREE | | 106.0 MB | 1.6 | 89.47 |
| | FASTXML | | 522.0 MB | 5.3 | 95.38 |
| | OVR | | 2.4 GB | - | 95.90 |
| IMAGENET | LTLS | | 0.16 MB | 15.0 | 0.75 |
| | W-LTLS | $b = 2$ | 0.16 MB | 21.4 | 0.42 |
| | | $b = 5$ | 0.34 MB | 26.3 | 1.59 |
| | | $b = 10$ | 0.84 MB | 40.3 | 4.36 |
| | | $b = 15$ | 1.2 MB | 52.6 | 4.08 |
| | | $b = 20$ | 1.8 MB | 76.2 | 7.01 |
| | | $b = 30$ | 3.7 MB | 194.2 | 12.60 |
| | LOMTREE | | 35.0 MB | 37.7 | 5.37 |
| | FASTXML | | 1.6 GB | 172.3 | 6.84 |
| | OVR | | 3.8 MB | - | 15.60 |
| DMOZ | LTLS | | 193.9 MB | 5.2 | 23.04 |
| | W-LTLS | $b = 2$ | 193.9 MB | 7.5 | 25.76 |
| | | $b = 3$ | 248.0 MB | 8.1 | 29.56 |
| | | $b = 5$ | 429.2 MB | 9.8 | 33.92 |
| | | $b = 10$ | 1.1 GB | 16.6 | 37.44 |
| | | $b = 20$ | 2.8 GB | 41.4 | 38.43 |
| | | $b = 30$ | 4.3 GB | 73.6 | 38.89 |
| | | $b = 40$ | 6.3 GB | 164.5 | 38.81 |
| | | $b = 50$ | 9.0 GB | 332.3 | 38.89 |
| | LOMTREE | | 1.8 GB | 28.0 | 21.27 |
| | FASTXML | | 1.2 GB | 60.7 | 38.58 |
| | OVR | | 38.0 GB | - | 35.50 |
| LSHTC1 | LTLS | | 256.3 MB | 0.65 | 9.50 |
| | W-LTLS | $b = 2$ | 256.3 MB | 1.1 | 10.56 |
| | | $b = 3$ | 361.6 MB | 1.1 | 13.72 |
| | | $b = 5$ | 631.6 MB | 1.3 | 16.58 |
| | | $b = 10$ | 1.5 GB | 2.2 | 20.44 |
| | | $b = 20$ | 4.0 GB | 5.1 | 21.76 |
| | | $b = 30$ | 6.3 GB | 9.9 | 22.08 |
| | | $b = 40$ | 9.0 GB | 20.4 | 22.30 |
| | | $b = 45$ | 10.8 GB | 29.3 | 22.47 |
| | LOMTREE | | 744.0 MB | 6.8 | 10.56 |
| | FASTXML | | 366.6 MB | 10.7 | 21.66 |
| | OVR | | 56.3 GB | - | 21.90 |

## E.6 Experimental results of the sparsity experiments

In the following Figure 11 we show that wider graphs induce models which use less features, even before pruning.

This may help understand the results in Section 7.3, where we show that the larger slice widths allow pruning more weights without accuracy degradation.

Figure 11: Percentage of non-zero weights at end of training (before pruning) vs the slice width $b$.

In the following Table 2, the results of Section 7.3 organized in a tabular form.

| DATASET | ALGORITHM | | | MODEL SIZE (BYTES) | TEST ACCURACY (%) |
|---|---|---|---|---|---|
| **SECTOR** | SPARSE W-LTLS | $b = 2$ | $\lambda = 0.324$ | 1.4 MB | 90.66 |
| | | $b = 4$ | $\lambda = 0.303$ | 1.0 MB | 93.42 |
| | | $b = 5$ | $\lambda = 0.327$ | 1.1 MB | 93.86 |
| | | $b = 7$ | $\lambda = 0.341$ | 1.0 MB | 94.11 |
| | | $b = 10$ | $\lambda = 0.358$ | 0.74 MB | 94.82 |
| | FASTXML | $T = 25$ | | 4.5 MB | 86.26 |
| | | $T = 50$ | | 8.9 MB | 86.26 |
| | | $T = 75$ | | 13.4 MB | 86.78 |
| | | $T = 100$ | | 17.9 MB | 86.58 |
| | | $T = 150$ | | 26.8 MB | 86.57 |
| | | $T = 200$ | | 35.8 MB | 87.20 |
| **ALOI.BIN** | SPARSE W-LTLS | $b = 2$ | $\lambda = 0.015$ | 37.3 MB | 84.09 |
| | | $b = 3$ | $\lambda = 0.015$ | 32.7 MB | 88.59 |
| | | $b = 5$ | $\lambda = 0.017$ | 25.4 MB | 91.74 |
| | | $b = 7$ | $\lambda = 0.016$ | 25.9 MB | 92.89 |
| | | $b = 10$ | $\lambda = 0.013$ | 30.3 MB | 94.48 |
| | | $b = 15$ | $\lambda = 0.014$ | 31.2 MB | 94.27 |
| | | $b = 20$ | $\lambda = 0.016$ | 25.4 MB | 94.55 |
| | FASTXML | $T = 5$ | | 52.1 MB | 92.67 |
| | | $T = 10$ | | 104.2 MB | 94.12 |
| | | $T = 25$ | | 260.9 MB | 95.19 |
| | | $T = 50$ | | 522.0 MB | 95.38 |
| | | $T = 100$ | | 1.0 GB | 95.66 |
| | DISMEC | | | 10.7 MB | 96.28 |
| | PD-SPARSE | | | 12.7 MB | 96.20 |
| | PPDSPARSE | | | 9.3 MB | 96.38 |
| **DMOZ** | SPARSE W-LTLS | $b = 2$ | $\lambda = 0.347$ | 43.2 MB | 24.64 |
| | | $b = 3$ | $\lambda = 0.325$ | 56.1 MB | 28.63 |
| | | $b = 5$ | $\lambda = 0.314$ | 69.3 MB | 33.04 |
| | | $b = 10$ | $\lambda = 0.323$ | 93.7 MB | 36.75 |
| | | $b = 20$ | $\lambda = 0.345$ | 126.3 MB | 37.72 |
| | | $b = 30$ | $\lambda = 0.385$ | 145.6 MB | 38.08 |
| | | $b = 40$ | $\lambda = 0.374$ | 193.1 MB | 38.08 |
| | | $b = 50$ | $\lambda = 0.323$ | 324.3 MB | 38.16 |
| | FASTXML | $T = 1$ | | 24.5 MB | 27.09 |
| | | $T = 2$ | | 49.2 MB | 31.17 |
| | | $T = 5$ | | 123.0 MB | 35.84 |
| | | $T = 10$ | | 246.5 MB | 37.47 |
| | | $T = 50$ | | 1.2 GB | 38.58 |
| | | $T = 300$ | | 7.4 GB | 38.63 |
| | DISMEC | | | 259.3 MB | 39.38 |
| | PD-SPARSE | | | 453.3 MB | 39.91 |
| | PPDSPARSE | | | 526.7 MB | 39.32 |
| **LSHTC1** | SPARSE W-LTLS | $b = 2$ | $\lambda = 0.288$ | 24.7 MB | 9.48 |
| | | $b = 3$ | $\lambda = 0.237$ | 27.6 MB | 12.66 |
| | | $b = 5$ | $\lambda = 0.271$ | 31.5 MB | 15.60 |
| | | $b = 10$ | $\lambda = 0.226$ | 46.3 MB | 19.78 |
| | | $b = 20$ | $\lambda = 0.207$ | 85.0 MB | 21.13 |
| | | $b = 30$ | $\lambda = 0.192$ | 130.5 MB | 21.56 |
| | | $b = 40$ | $\lambda = 0.183$ | 184.7 MB | 21.81 |
| | | $b = 45$ | $\lambda = 0.246$ | 152.1 MB | 21.88 |
| | FASTXML | $T = 1$ | | 7.3 MB | 12.50 |
| | | $T = 5$ | | 36.6 MB | 17.98 |
| | | $T = 10$ | | 73.3 MB | 19.36 |
| | | $T = 50$ | | 366.6 MB | 21.66 |
| | | $T = 150$ | | 1.1 GB | 21.94 |
| | | $T = 300$ | | 2.2 GB | 22.04 |
| | | $T = 1200$ | | 8.8 GB | 21.92 |
| | DISMEC | | | 94.7 MB | 22.74 |
| | PD-SPARSE | | | 58.7 MB | 22.46 |
| | PPDSPARSE | | | 254.0 MB | 22.70 |

Table 2: Simulation results for the sparse models.

## Footnotes

[5] The model size is linear in the number of predictors, which in turn depends on the slice width like $\frac{b^2}{\log b}$.