[Reviews · NeurIPS 2018]

Reviewer 1



This paper proposes an algorithm to solve extreme multi-class classification problems using Error Correcting Output Coding. During training, the algorithm simply learns l (logK) independent binary classifiers. Main contribution of the paper is in the inference algorithm. It reduces the costly loss based decoding framework for ECOC to that of finding shortest path on a weighted trellis graph. The paper is well written and easy to understand. The proposed algorithm is quite interesting and novel. However, I have a few reservations - 1) Authors should have also compared their algorithm against a recently(WWW2018) proposed extreme classification algorithm called Parabel [1]. Parabel gives same accuracy as one-vs-rest(OVR) but is orders of magnitude faster than OVR both during training and prediction. Moreover, its model size is also much smaller as compared to FastXML. 2) As a baseline comparison it might be better to include the results with Hamming Decoding as well (keeping the training procedure same as LTLS) 3) As shown in DiSMEC paper[2] the model size of OVR can be significantly reduced by simple hard thresholding. So, authors should have reported the model size of OVR after hard thresholding. Often this model size comes out to be much less than FastXML(ref. [1][2]) and hence might beat the proposed algorithm in terms of model size. Simply dismissing the comparison with methods like DiSMEC and PPDSparse by arguing that “line of research is orthogonal to the ECOC approach, which can also employ sparse models to reduce the model size ” does not seem fair, particularly because all the datasets considered in the paper have hight dimensional sparse bag-of-words features. 4) To improve readability of the paper, instead of just providing the comparison graphs, it might be better if results are also reported in a table. 5) In Accuracy vs prediction time/model size graphs for fastXML just a single point is plotted. For fastXML prediction time/model size can be easily varied by changing the number of trees. So, authors should have presented the entire curve as is done for W-LTLS 6) It might be useful to include some discussion on training time of W-LTLS. I am assuming it would be orders of magnitude faster to train as compared to OVR which gives the best prediction accuracy. [1]Y. Prabhu, A. Kag, S. Harsola, R. Agrawal and M. Varma. Parabel: Partitioned label trees for extreme classification with application to dynamic search advertising. In Proceedings of the ACM International World Wide Web Conference, Lyon, France, April 2018. [2]R. Babbar, and B. Schölkopf, DiSMEC - Distributed Sparse Machines for Extreme Multi-label Classification in WSDM, 2017

Reviewer 2



1. Summary The paper concerns a problem of extreme classification. The authors provide an in-depth analysis of the recently proposed LTLS algorithm. They reformulate the problem in the framework of loss-based decoding for ECOC. Based on their theoretical insights they introduce valid extensions to the original algorithm. They discuss the trade-off between the computational and predictive performance, which is then validated experimentally. The introduced W-LTLS obtains promising results in the experiments. 2. Relation to previous work The authors build their work on the LTLS algorithm and the loss-based decoding for ECOC. Both are very nicely introduced in a formal way at the beginning of the paper. The authors also include a section with a more general overview of related work. Below I give only one minor remark in this regard. Since the authors also discuss the tree-based methods in the related work, the following algorithms (with logarithmic complexity) should be mentioned: - Hierarchical softmax ("Hierarchical probabilistic neural network language model", AISTATS 2005) and FastText ("Bag of tricks for efficient text classification", CoRR 2016) - Conditional probability estimation trees ("Conditional probability tree estimation analysis and algorithms", UAI 2009) - Classification Probability Trees (the generalization of the above) ("Consistency of probabilistic classifier trees", ECMLPKDD 2016) - Filter trees ("Error- correcting tournaments", ALT 2009) - Probabilistic label trees ("Extreme F-measure maximization using sparse probability estimates", ICML 2016) 3. Quality: 3.1 Strengths This is a solid submission that extends previous work. The new algorithm has been analyses from theoretical and empirical point of view. The authors also try to establish in a formal way the trade-off between predictive accuracy and computational complexity. This is one of the first steps into a general reduction framework for this kind of trade-off. 4. Clarity 4.1 Strengths The paper is well and clearly written. The appendix contains almost all necessary elements for which there were no place in the main text. 4.2 Weaknesses The empirical results should also be given in tabular form with numeric values, not only in the form of plots. This is important for accurate comparison of algorithms in the future references. 5. Originality: 5.1 Strengths The algorithm and the theoretical results are original. 6. Significance: 6.1 Strengths The work is significant as it will push the research in the desired direction of establishing the framework for the trade-off between predictive and computation performance. 7. Minor remarks: References: Reference 1 has been doubled, please check also other references as they do not contain all relevant information. 8. After rebuttal I thank the authors for their rebuttal. I do not change my evaluation, but give one additional comment that could be useful for the authors. As the paper compares W-LTLS to LomTree and FastXML, it could also include the comparison to a label tree approach being either represented by HSM, PLT, or Parabel. All these algorithms are very similar to each other. PLT and Parabel are multi-label generalizations of HSM. Parabel additionally uses a specific tree building strategy that gets very good results for multi-label problems. To compress the model size in label tree approaches one can use a simple shallow network as it is done in the FastText implementation.

Reviewer 3



The paper is clearly explaining the problem they are solving and makes good connections with OneVersusRest and LTLS, such as how they cast the previous LTLS work as an instance of loss based decoding with squared loss. Wide-LTLS is an interesting way to extend LTLS. I was hoping the authors could address increasing the Hamming distance. I am not sure the Hamming distance is 4 as claimed because there are eventually one path per class so I would expect the Hamming distance to be 1. The experiments are extensive but I am surprised timings are not very competitve. I would expect this method to be much faster than the baselines.